# Spatial Distribution of Butterflies in Accordance with Climate Change in the Korean Peninsula

**Sangdon Lee ***[ID]**, Hyeyoung Jeon and Minkyung Kim**

Department of Environmental Sciences & Engineering, College of Engineering, Ewha Womans University, Seoul 03760, Korea; hyth4197@hanmail.net (H.J.); pollyfriend@nate.com (M.K.)

* Correspondence: lsd@ewha.ac.kr

**Abstract:** The effects of climate change are becoming apparent in the biosphere. In the 20th century, South Korea experienced a 1.5 °C temperature increase due to rapid industrialization and urbanization. If the changes continue, it is predicted that approximately 15–37% of animal and plant species will be endangered after 2050. Because butterflies act as a good indicator for changes in the temperature, the distribution of butterflies can be used to determine their adaptability to climate patterns. Local meteorological data for the period 1938–2011 were used from the National Forest Research Institute of Korea. Local temperature data were additionally considered among the basic information, and the distribution patterns of butterflies were analyzed for both the southern and northern regions. Southern butterflies (with northern limit) tend to increase in number with significant correlation between the temperature and number of habitats ($p < 0.000$), while northern butterflies (with southern limit) show no statistical significance between the temperature and number of habitats, indicating their sensitivity to temperature change. This finding is in accordance with the conclusion that southern butterflies are more susceptible to climate change when adapting to local environments and expanding their original temperature range for survival, which leads to an increase in the numbers of their habitats.

**Keywords:** butterflies; global warming; habitat shift; spatial distribution; northern species

---

## 1. Introduction

Industrialization and urbanization are leading to global warming problems that are causing the Earth's temperature to rise rapidly. The effects of this climate change are apparent in the biosphere [1]; thousands of species are migrating toward suitably adapted habitats (area of occupation). Changes in habitat range are actively progressing because of decreased climate-compatible habitats and increased risk of species extinction [1], which are key examples of the risks posed by climate change [2].

Among all living things, insects are sensitive to temperature changes and, of this group, butterflies are useful indicators of climate change; they are easy to examine, well known for their life cycles, and sensitive to the environment [3–5]. Climate change has the potential to seriously affect butterfly populations and has been linked to mass mortality at overwintering sites, population range shifts, and extirpation from fluctuating precipitation levels [6].

Recently, climate change research has been actively conducted on butterflies. In the Northern Hemisphere, 35 species of butterflies moved up 35 to 240 km due to climate change [5], while research into Australian climatic scenarios has shown that more than 80% of unique species are expected to disappear by 2050 [7].

Furthermore, butterfly activity has become rapid with increasing temperature in Britain [8], Spain [9], and North America [10]. In Japan, the great Mormon (*Papilio memnon*) and red Helen (*P. helenus*) species, found in the south, are expanding their distributions [11].

---

Recently, butterfly research in South Korea has been carried out against the backdrop of climate change. Kwon et al. [12] found that the some of the existing northern groups had decreased in number. Most of these studies have focused on populations, and although data on the changes in the distribution of butterflies on the Korean Peninsula are available, few studies on their relationship with the temperature have been conducted. In the current study, the researcher identified the changes in the butterfly population in the Korean Peninsula over 73 years, from 1938 to 2011. It is significant that the entire region of the Korean Peninsula was surveyed and analyzed for this period (1938–2011) for butterfly distribution. In addition, distribution changes were analyzed using local temperature data, and differences between the southern and northern regions were studied, taking into consideration the distribution patterns and ecological characteristics.

The aim of this work was to identify the changes in the distribution of butterfly habitats as a result of climate change: (1) To correlate temperature and the number of habitats (with the presence of butterflies) for the southern and northern species based on their distribution patterns, and (2) to analyze the differences in the distribution of habitats according to the latitude and to discover changes in the distribution patterns by period.

This study will serve as a basis for the changes in butterfly distribution patterns due to climate change and will serve as a guideline on providing a management plan for butterfly species, helping predict the later disappearance or survival of species.

## 2. Methods and Study Areas

### 2.1. Meteorological Data

The nation's weather forecasts began as early as 1904 with Incheon (Station No. 112), and the total number of weather stations was 79 (Figure 1, Appendix A, Table A1). In this study, weather station data were used to calculate the average temperature per cell grid (habitat) considering the period in which the meteorological observatory began and ended.

The overall period was divided into four segments—1938–1955, 1956–1975, 1976–1996, and 1997–2011—according to the availability of data. The average temperature per period was calculated by averaging the periods following the annual average temperature calculations.

### 2.2. Butterfly Distribution Data

The book "Changing Distribution of Butterflies in Korea" [12] was used as a reference that provided the basic data, as it had compiled all records of butterfly distribution for the period 1938–2011 including the studies of Seok [13] (1938–1955), Kim [14] (1955–1975), Park and Kim [15] (1977–1996), and Kim and Seo [16] (1996–2011). The standard method was used to collect data on in-line transect methods (30 paces/min) and observed butterflies within 10 m of both in-line transects. Butterfly species composition and relative abundances were sampled using transect counts, modified from the method proposed by Pollard and Yates (1993) [17]. Even though the Pollard–Walk method did not exist before the 1970s, the observation was conducted in a manner similar to the line transect method assuming standardized collection of data and quality of data. All butterflies seen within bounds of route (5 m width recorded) and within 5 m ahead were recorded.

Observations were made when butterflies appeared (March to November). In an early publication, Seok [13] showed butterfly appearances by location and later converted them to GPS points so that if one observed the species we created "presence" in each cell. A total of 255 species were observed, which were grouped into three: Southern (with northern limit), Northern (with southern limit), and Miscellaneous (Appendix B). Northern species were defined as species for which the southern boundary in East Asia is located within the Korean peninsula whereas the southern species had a northern boundary of being observed more often in southern areas than northern areas [4]. Butterfly species not classified as Northern or Sothern were defined as "Miscellaneous" species.

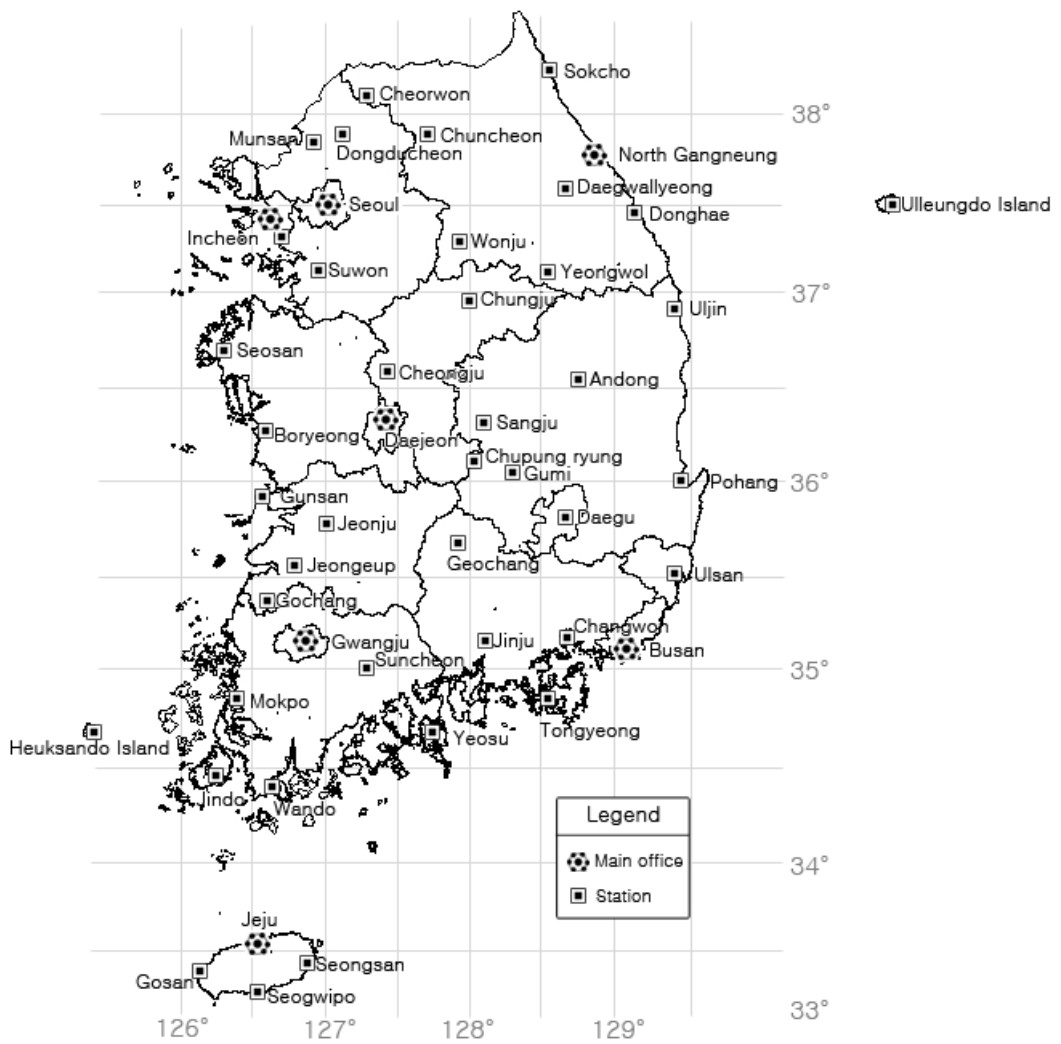

**Figure 1.** Meteorological observation network operating in Korean Regional Meteorological Office (There are five main offices—Seoul, Noyth Gangneung, Da Daejeon, Gwangju, Busan—and 75 stations).

Cases with no clear limit species frequencies were considered. The number of grid changes over time was indicated for 181 different species of butterfly (out of 225) in Korea that could be analyzed. The grid cell was created on latitude 0.5° (56 km) × longitude 0.5° (44.4 km), and observation was marked in the grid cell, and a total of 99 grid cells were created (9 for longitude, 11 for latitude). In this work, the scope of the data was based on land areas excluding marine areas, and grid species (Cell) were counted using the number of butterflies. The grid-specific temperature data was further considered using the grid and the weather station data therein. The cell grids were used to represent the species "area of occupation" (i.e., habitat) according to the temperature.

### 2.3. Data Analysis

ANOVA was applied to determine how the temperatures influence the habitat shift, and Tukey HSD, which is applicable for pairwise comparison of means, was applied post-hoc to monitor change in distribution during the four time periods [18]. The correlation between the temperature and butterfly distribution was applied using linear regression with the help of SPSS (IBM, New York, NY, USA, version 21.0).

## 3. Results

### 3.1. Change by Periods

During the study periods, there was an increase in annual average temperature according to latitudes (Table 1). The temperature changes by latitude were analyzed and divided into four time segments (<35°, 35° to 36°, 36° to 37°, and >37°) to be determined. For all periods, it was observed that the higher the latitude, the lower the temperature. This also indicated an increase in the latitude temperature over time.

**Table 1.** Change in annual average temperature according to latitude.

| Period / Latitude | Temperature | | | |
|---|---|---|---|---|
| | **1938–1955** | **1955–1975** | **1977–1996** | **1996–2011** |
| Over 37° | 11.4 | 11.5 | 11.5 | 11.9 |
| 36–37° | 11.6 | 11.7 | 12.0 | 12.4 |
| 35–36° | 12.9 | 13.2 | 13.3 | 13.9 |
| Under 35° | 13.8 | 14.0 | 14.3 | 14.6 |

Temperature changes were investigated by selecting the representative major regions to identify seasonal changes in temperatures in the Korean Peninsula. The warmest regions were the southern regions, such as Jeju Island, while the middle and northern regions showed lower temperatures. As time went by, the temperature tended to increase (Figure 2).

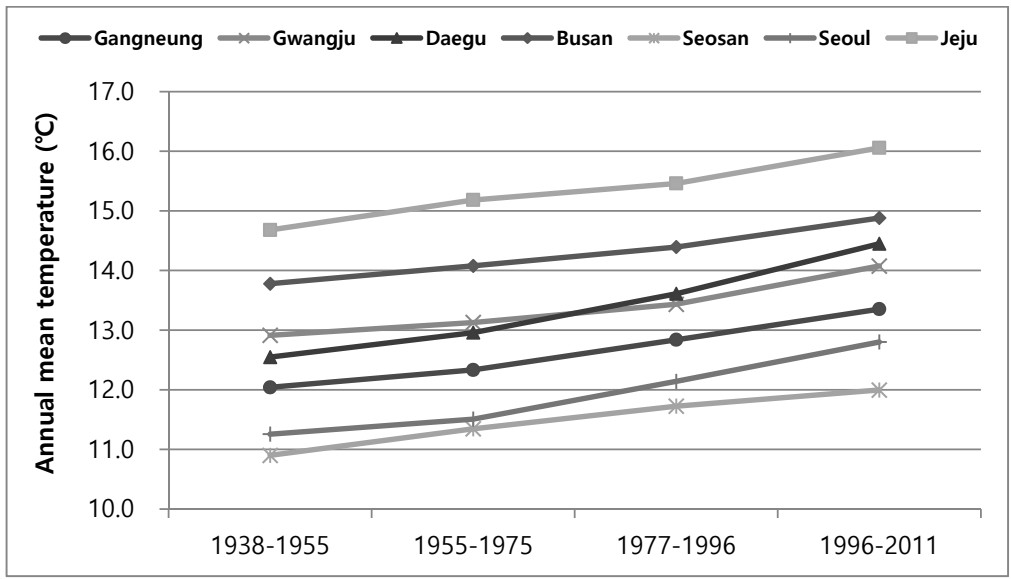

**Figure 2.** Periodical temperature change in Korean seven major cities.

An ANOVA was conducted to determine whether the habitat temperatures of the southern and northern areas were different among the time periods ($p < 0.05$) (Table 2). The southern region increased in average temperatures over time, and the standard deviation of 1955–1975 was the highest among all periods (12.6 ± 77), showing a large variation in temperature during these periods.

Box plots were applied to identify changes in the historical temperature range. Both the southern and the northern species temperatures became higher as time went by, and the trends were more apparent in southern regions (Figure 3). Looking at the number of seasonal habitats for all butterflies, this number gradually increased over time (Figure 4), except in 1955–1975. Periodic changes in the cell grids in southern and northern butterflies showed that northern species were higher in terms of number of habitats (Figure 3).

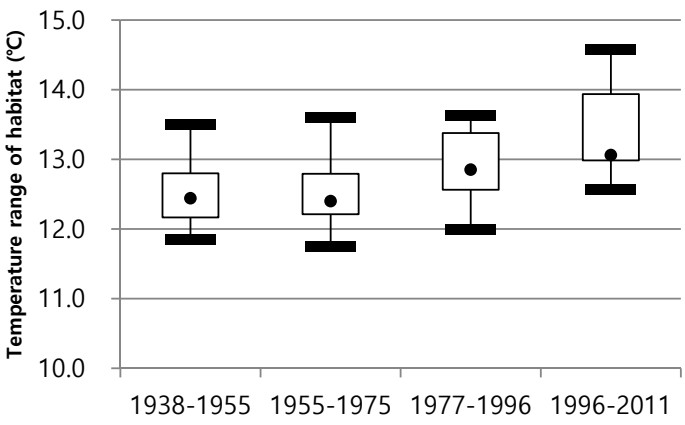

Southern species of butterflies

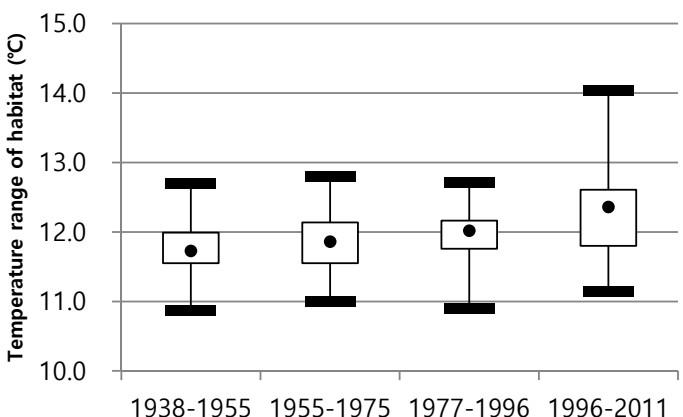

Northern species of butterflies

**Figure 3.** Temperature range of habitat with time periods in southern and northern species during 1938–2011. Box plot of periodical temperature range change in Southern and Northern butterflies (upper bar: 75%, lower bar: 25%, •: median). Both cases showed $p < 0.05$.

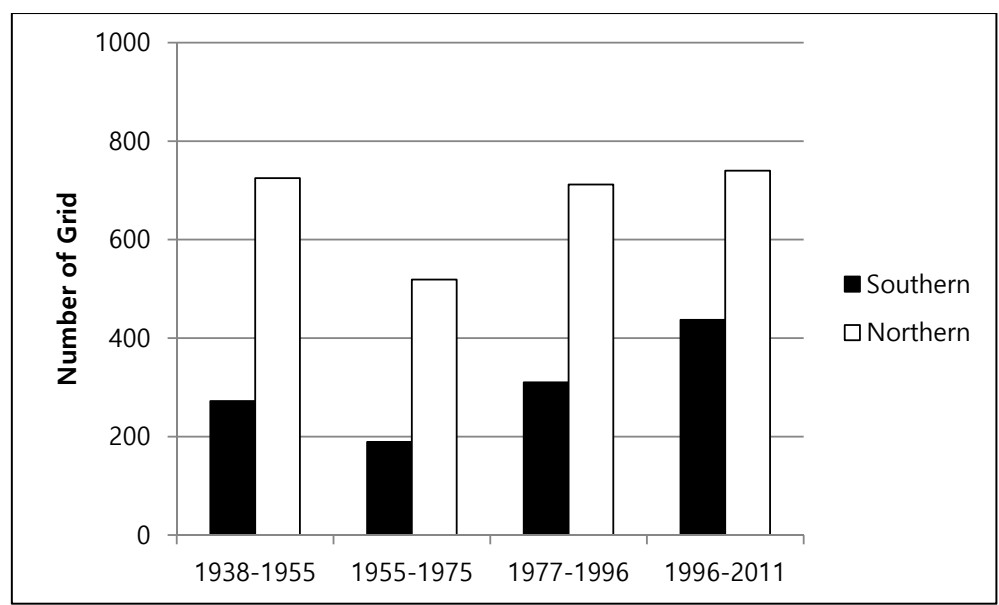

**Figure 4.** Periodical change in habitat number grids in Southern and Northern butterflies.

**Table 2.** Temperature changes in Southern and Northern butterflies.

| Distribution Pattern | SS | DF | F | *p* |
|---|---|---|---|---|
| Southern | 6.965 | 3 | 6.624 | 0.001 * |
| Northern | 8.330 | 3 | 15.419 | 0.000 * |

* *p* < 0.05.

### 3.2. Change in Number of Habitats According to Temperature

The number of habitats showed different trends: The habitat numbers were classified into four latitudes (<35°, 35° to 36°, 36° to 37°, and >37°) during each period (1938–1955; 1956–1975; 1976–1996, and 1997–2011). A two-way ANOVA was applied for southern and northern species. There was no significance for the southern species; however, for the northern groups, there was significance for latitude (*p* < 0.00003) and year (*p* < 0.02) (Tables 3 and 4), indicating that southern species tend to expand their territories with increasing temperature (Figure 5).

**Table 3.** Habitat number and percentage (%) according to latitude in Southern butterflies.

| Period / Latitude | 1938–1955 # of Grids | (%) | 1955–1975 # of Grids | (%) | 1977–1996 # of Grids | (%) | 1996–2011 # of Grids | (%) |
|---|---|---|---|---|---|---|---|---|
| Over 37° | 78 | 26.2 | 53 | 30.8 | 89 | 30.8 | 136 | 32.2 |
| 36–37° | 72 | 24.2 | 43 | 25.0 | 49 | 17.0 | 90 | 21.3 |
| 35–36° | 92 | 30.9 | 50 | 29.1 | 75 | 26.0 | 109 | 25.8 |
| Under 35° | 56 | 18.8 | 26 | 15.1 | 76 | 26.3 | 87 | 20.6 |
| Total | 298 | 100.0 | 172 | 100.0 | 289 | 100.0 | 422 | 100.0 |

**Table 4.** Habitat number and percentage (%) according to latitude in Northern butterflies.

| Period / Latitude | 1938–1955 # of Grids | (%) | 1955–1975 # of Grids | (%) | 1977–1996 # of Grids | (%) | 1996–2011 # of Grids | (%) |
|---|---|---|---|---|---|---|---|---|
| Over 37° | 396 | 54.6 | 265 | 52.6 | 445 | 62.5 | 463 | 62.6 |
| 36–37° | 141 | 19.4 | 103 | 20.4 | 92 | 12.9 | 122 | 16.5 |
| 35–36° | 161 | 22.2 | 107 | 21.2 | 136 | 19.1 | 113 | 15.3 |
| Under 35° | 27 | 3.7 | 29 | 5.8 | 39 | 5.5 | 42 | 5.7 |
| Total | 725 | 100.0 | 504 | 100.0 | 712 | 100.0 | 740 | 100.0 |

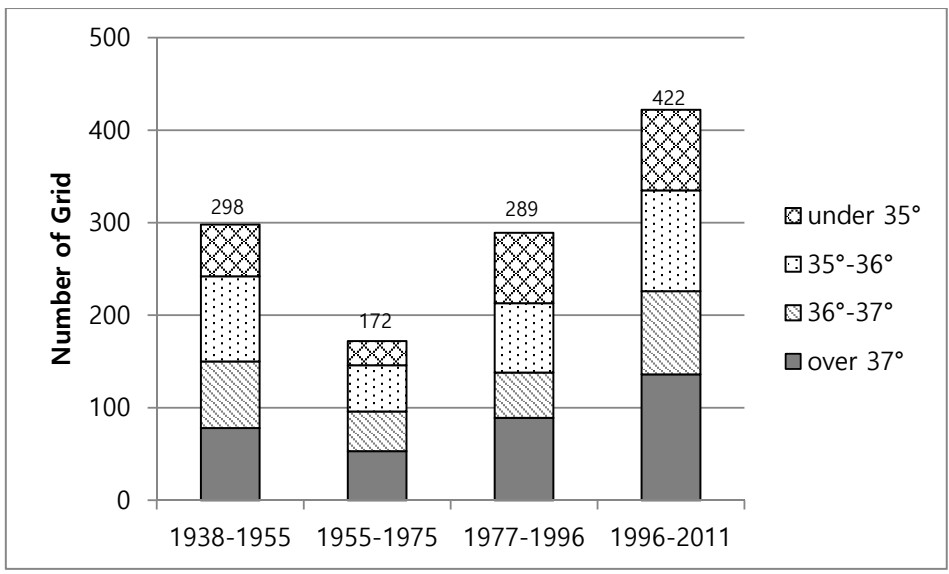

**Figure 5.** Change in habitat number according to latitude in Southern butterflies.

The number of habitat cells was increasing in higher latitudes (over 37°) for southern butterflies, indicating a northward shift of habitats due to climate change. Findings in species richness showed similar results: The higher the latitude, the greater the richness of species in both groups. According to correlation analysis between latitude and number of species, northern butterflies showed higher significance ($p < 0.001$), and number of species was greatly affected by the latitude. Considering their sensitiveness to temperature, it can be assumed that northern butterflies' species richness and number of habitats will decrease as their habitat becomes less suitable for them. As for southern butterflies, we may expect a sizable increase in both species' richness and number of habitats because climate change has made South Korea a habitable area for them.

The number of habitats varied among periods, with Tukey HSD results showing that for southern species, the number of habitats was different except in 1977–1996, while for northern species it varied except during 1938–1955 and 1977–1996 (Table 5).

**Table 5.** Multiple comparison results between number of habitat data using HSD analysis (post-hoc) in Southern and Northern butterfly data.

| Distribution Pattern | (I) Period | (J) Period | (I)–(J) | *p* |
|---|---|---|---|---|
| Southern | 1938–1955 | 1955–1975 | 4.882 | 0.444 |
| | | 1977–1996 | −2.235 | 0.902 |
| | | 1996–2011 | −9.706 | 0.021 * |
| | 1955–1975 | 1938–1955 | −4.882 | 0.444 |
| | | 1977–1996 | −7.118 | 0.138 |
| | | 1996–2011 | −14.588 | 0.000 * |
| | 1977–1996 | 1938–1955 | 2.235 | 0.902 |
| | | 1955–1975 | 7.118 | 0.138 |
| | | 1996–2011 | −7.471 | 0.110 |
| | 1996–2011 | 1938–1955 | 9.706 | 0.021 * |
| | | 1955–1975 | 14.588 | 0.000 * |
| | | 1977–1996 | 7.471 | 0.110 |
| Northern | 1938–1955 | 1955–1975 | 0.866 | 0.878 |
| | | 1977–1996 | −2.015 | 0.307 |
| | | 1996–2011 | −2.433 | 0.157 |
| | 1955–1975 | 1938–1955 | −0.866 | 0.878 |
| | | 1977–1996 | −2.881 | 0.065 |
| | | 1996–2011 | −3.299 | 0.025 * |
| | 1977–1996 | 1938–1955 | 2.015 | 0.307 |
| | | 1955–1975 | 2.881 | 0.065 |
| | | 1996–2011 | −0.418 | 0.984 |
| | 1996–2011 | 1938–1955 | 2.433 | 0.157 |
| | | 1955–1975 | 3.299 | 0.025 * |
| | | 1977–1996 | 0.418 | 0.984 |

* $p < 0.05$.

## 4. Discussion

Many studies have already shown that butterflies are among species that have responded the most to climate change, usually in the form of northward or elevation range shifts [19]. Temperatures in the Korean peninsula have increased rapidly since the 1960s due to rapid industrialization and urbanization. The average temperatures for the last period (1996–2011) were higher than those in the initial period (1938–1955) in Gangneung (1.4 °C), Seoul (1.5 °C), and Jeju Island (1.5 °C), with temperatures in most locations increasing by over 1.0 °C on average.

Both the southern and northern communities have seen an increase in temperature such that butterflies must adapt to the local temperatures as the climate changes. In addition, the southern parts had a higher temperature of habitat than the northern areas such that species groups in the southern regions showed an increase in habitat numbers over time (Table 5). It was found that as the temperature increases, the species in southern regions are more sensitive to temperature so that they tend to expand their territories in the face of climate change. Kwon et al. [20] also indicated that southern species tend to expand their territories to the north, meaning that increasing temperature could be an important factor for a habitat shift.

As climate change and global warming increase, many species are also adapting to their local environmental conditions so that a changing distribution will be seen depending on their adaptability [1]. Parmesan [21] indicated that butterflies living at low latitudes moved slowly northward, greatly increasing at higher latitudes. Species in the southern communities tended to move northward, most to temperatures higher than 37 °C, and the same result was found not only in Korea, but also in Britain and Europe [22].

The northern region has a larger temperature range than the south, indicating that climate change has significantly affected the distribution patterns of butterflies, especially during 1996–2011. Habitat shifts in both areas showed significance ($p < 0.05$), indicating that both northern and southern species are sensitive to temperature (Table 5). Choi [23] also demonstrated that species richness at northern altitudes should be increasing due to global warming and species' adaptability to warming temperature. The Korean butterflies were divided into two groups of Palearctic species coming from the continent and Oriental species migrating across the ocean, indicating that northern species from the Palearctic have a chance to expand their habitat due to warming temperature, a trend that was observed in this study [23,24].

An overall decline occurred during 1956–75 due to habitat destruction after the Korean war and the rapid expansion of urbanization in the 1960s and 1970s [25,26]. Artificial factors, such as war, presumably can be important factors that influence the anemogram of species [27]. Artificial disturbances such as temperature changes and wars have a direct impact on the habitat of butterflies and their population, population structure, and species abundance.

Ecological status should be based on both biotic and physical environmental factors. Pianka [28] indicated that butterflies should have their own ecological status depending on the changing environment. The southern and northern species differ in based on their adaptability to temperature zones. The southern species, which are mostly located in the southern regions, have sensitivity to relatively high habitat temperature, while the northern species had a cooler temperature than their southern counterparts.

Why are southern species so sensitive to warming temperature, showing a greater habitat shift than their northern neighbors? Climate change can affect flight times in butterflies. Warmer temperatures will result in more generations of multiple–brooded species, but how this will affect egg-laying periods and other life traits determined by photoperiod (due to climate change) is unknown [29]. However, this study showed the general patterns of southern species expanding their territory to the north. Disease can also harm butterfly populations, with recent studies suggesting that populations whose migration is at risk may be even more susceptible to outbreaks of disease [30,31]. Habitat loss and fragmentation can lead to population declines and local extinctions [32], and the use of herbicides on crops can reduce host and nectar plant availability in agricultural settings [33–35]. However, why the habitat in the north was more significant should be answered with the help of GIS or other techniques that imply spatial analysis of habitat.

As was the case with the studies by Pollard et al. [36,37], Warren et al. [22] in the UK, and Hill et al. [38] in the EU, it was found that butterfly species have gradually extended north as climate changes continue. On the other hand, a study by Parmesan et al. [5] showed that a small proportion of butterflies migrate to lower latitudes. These results are consistent with the finding that most southern species move to upper latitudes when expanding their territory.

In this study, the researcher identified the overview of the Korean Peninsula's butterfly transformation over the 73 years from 1938 to 2011. It is significant that the entire length of this time for butterfly distribution was analyzed on the Korean Peninsula. Additionally, changes in distribution patterns were analyzed by giving consideration to the temperature, by constructing the local temperature data. The distribution characteristics of the southern and northern areas due to temperature changes can be used in various conservation strategies for butterfly populations. If such changes are confirmed, the forecast for the change in the population density can be made together with the change in the weather.

**Author Contributions:** Conceptualization, writing and editing—original draft preparation and supervision, funding acquisition, S.L.; data analysis, H.J.; validation and literature review, M.K. All authors have read and agreed to the published version of the manuscript.

**Funding:** This research was funded by Basic Science Research Program through the National Research Foundation of Korea (NRF) funded by the Ministry of Science, Technology and Education (NRF–2017R1D1A1B03029300) and Environmental Science & Technology Center (SEST), KOREA, and SEST (2020).

**Conflicts of Interest:** No conflict of interests among authors.

**Appendix A**

**Table A1.** Meteorological stations operating in Korean Regional Meteorological Office.

|  | **Station** | **Lat. (N)** | **Long (E)** | **H (m)** | **Hb (m)** | **Ht (m)** | **Ha (m)** | **Hr (m)** |
|---|---|---|---|---|---|---|---|---|
| 90 | Sokcho | 38°15′ | 128°33′ | 18.1 | 24.3 | 1.9 | 10.0 | 0.7 |
| 95 | Cheorwon | 38°08′ | 127°18′ | 153.7 | 156.4 | 1.8 | 12.6 | 0.6 |
| 98 | Dongducheon | 37°54′ | 127°03′ | 109.1 | 113.6 | 1.7 | 10.0 | 0.6 |
| 99 | Paju | 37°53′ | 126°45′ | 29.4 | 31.4 | 1.7 | 10.0 | 0.5 |
| 100 | Daegwallyeong | 37°40′ | 128°43′ | 772.6 | 773.7 | 1.8 | 10.0 | 0.6 |
| 101 | Chuncheon | 37°54′ | 127°44′ | 77.7 | 77.8 | 1.5 | 10.0 | 0.6 |
| 102 | Baengnyeongdo | 37°57′ | 124°37′ | 144.9 | 146.6 | 1.8 | 9.4 | 0.6 |
| 104 | Bukgangneung | 37°48′ | 128°51′ | 78.9 | 80.3 | 1.6 | 10.0 | 0.5 |
| 105 | Gangneung | 37°45′ | 128°53′ | 26.0 | 27.5 | 1.7 | 17.9 | 0.6 |
| 106 | Donghae | 37°30′ | 129°07′ | 39.9 | 40.6 | 1.7 | 10.0 | 0.6 |
| 108 | Seoul | 37°34′ | 126°57′ | 85.8 | 86.5 | 1.5 | 10.0 | 0.6 |
| 112 | Incheon | 37°28′ | 126°37′ | 71.4 | 73.4 | 1.5 | 10.0 | 1.7 |
| 114 | Wonju | 37°20′ | 127°56′ | 148.6 | 152.2 | 1.6 | 10.0 | 0.6 |
| 115 | Ulleungdo | 37°28′ | 130°53′ | 222.8 | 224.1 | 1.8 | 10.0 | 0.6 |
| 119 | Suwon | 37°16′ | 126°59′ | 34.1 | 35.5 | 1.5 | 18.7 | 0.5 |
| 121 | Yeongwol | 37°10′ | 128°27′ | 240.6 | 240.7 | 1.5 | 10.0 | 0.6 |
| 127 | Chungju | 36°58′ | 127°57′ | 115.1 | 117.7 | 1.8 | 10.0 | 0.5 |
| 129 | Seosan | 36°46′ | 126°29′ | 28.9 | 29.9 | 1.3 | 20.2 | 0.6 |
| 130 | Uljin | 36°59′ | 129°24′ | 50.0 | 50.6 | 1.8 | 13.0 | 0.6 |
| 131 | Cheongju | 36°38′ | 127°26′ | 57.2 | 57.9 | 1.5 | 10.0 | 0.5 |
| 133 | Daejeon | 36°22′ | 127°22′ | 68.9 | 70.1 | 1.6 | 19.8 | 0.6 |
| 135 | Chupungnyeong | 36°13′ | 127°59′ | 244.7 | 246.0 | 1.5 | 10.0 | 0.6 |
| 136 | Andong | 36°34′ | 128°42′ | 140.1 | 142.1 | 1.7 | 10.0 | 0.6 |
| 137 | Sangju | 36°24′ | 128°09′ | 96.2 | 99.4 | 1.6 | 10.0 | 0.5 |
| 138 | Pohang | 36°01′ | 129°22′ | 2.3 | 2.7 | 1.6 | 15.4 | 0.6 |
| 140 | Gunsan | 36°00′ | 126°45′ | 23.2 | 28.3 | 1.7 | 15.3 | 0.6 |
| 143 | Daegu | 35°53′ | 128°37′ | 64.1 | 65.2 | 1.8 | 10.0 | 0.6 |
| 146 | Jeonju | 35°49′ | 127°09′ | 53.4 | 62.4 | 1.8 | 18.4 | 0.6 |
| 152 | Ulsan | 35°33′ | 129°19′ | 34.6 | 35.8 | 1.5 | 12.0 | 0.5 |
| 155 | Changwon | 35°10′ | 128°34′ | 37.2 | 37.9 | 1.7 | 10.0 | 0.5 |
| 156 | Gwangju | 35°10′ | 126°53′ | 72.4 | 75.3 | 1.5 | 17.5 | 0.6 |
| 159 | Busan | 35°06′ | 129°01′ | 69.6 | 70.2 | 1.6 | 17.8 | 0.6 |
| 162 | Tongyeong | 34°50′ | 128°26′ | 32.7 | 33.7 | 1.5 | 15.2 | 0.6 |
| 165 | Mokpo | 34°49′ | 126°22′ | 38.0 | 38.6 | 1.5 | 15.5 | 0.6 |

**Table A1.** *Cont.*

| | Station | Lat. (N) | Long (E) | H (m) | Hb (m) | Ht (m) | Ha (m) | Hr (m) |
|---|---|---|---|---|---|---|---|---|
| 168 | Yeosu | 34°44′ | 127°44′ | 64.6 | 74.6 | 1.5 | 20.8 | 0.6 |
| 169 | Heuksando | 34°41′ | 125°27′ | 76.5 | 77.9 | 1.7 | 9.0 | 0.6 |
| 170 | Wando | 34°23′ | 126°42′ | 35.2 | 28.4 | 1.6 | 15.4 | 0.5 |
| 172 | Gochang | 35°20′ | 126°35′ | 52.0 | 53.2 | 1.5 | 10.0 | 1.7 |
| 174 | Suncheon | 35°01′ | 127°22′ | 165.0 | 180.4 | 1.8 | 10.3 | 0.6 |
| 175 | Jindo | 34°28′ | 126°19′ | 476.5 | 477.8 | 1.6 | 10.0 | 0.5 |
| 176 | Daegu | 35°52′ | 128°39′ | 49.0 | 50.2 | 1.8 | 10.0 | 0.6 |
| 184 | Jeju | 33°30′ | 126°31′ | 20.4 | 21.1 | 1.8 | 12.3 | 0.6 |
| 185 | Gosan | 33°17′ | 126°09′ | 74.3 | 75.6 | 1.8 | 10.0 | 0.6 |
| 188 | Seongsan | 33°23′ | 126°52′ | 17.8 | 20.1 | 1.5 | 10.0 | 0.6 |
| 189 | Seogwipo | 33°14′ | 126°33′ | 49.0 | 50.2 | 1.9 | 10.0 | 0.6 |
| 192 | Jinju | 35°09′ | 128°02′ | 30.2 | 31.5 | 1.5 | 10.0 | 0.7 |
| 201 | Ganghwa | 37°42′ | 126°26′ | 47.0 | 47.3 | 1.6 | 12.0 | 0.6 |
| 202 | Yangpyeong | 37°29′ | 127°29′ | 48.0 | 48.6 | 1.7 | 10.0 | 0.6 |
| 203 | Icheon | 37°15′ | 127°29′ | 78.0 | 91.0 | 1.9 | 10.0 | 0.5 |
| 211 | Inje | 38°03′ | 128°10′ | 200.2 | 201.5 | 1.5 | 10.0 | 0.5 |
| 212 | Hongcheon | 37°41′ | 127°52′ | 140.9 | 147.2 | 1.6 | 13.0 | 0.5 |
| 216 | Taebaek | 37°10′ | 128°59′ | 712.8 | 715.3 | 1.7 | 16.0 | 0.6 |
| 221 | Jecheon | 37°09′ | 128°11′ | 263.6 | 263.9 | 1.5 | 13.3 | 0.5 |
| 226 | Boeun | 36°29′ | 127°44′ | 175.0 | 176.4 | 1.5 | 10.0 | 0.5 |
| 232 | Cheonan | 36°46′ | 127°07′ | 21.3 | 22.6 | 1.8 | 9.5 | 0.6 |
| 235 | Boryeong | 36°19′ | 126°33′ | 15.5 | 18.9 | 1.6 | 9.8 | 0.5 |
| 236 | Buyeo | 36°16′ | 126°55′ | 11.3 | 12.3 | 1.7 | 9.5 | 0.5 |
| 238 | Geumsan | 36°06′ | 127°28′ | 170.4 | 171.6 | 1.5 | 10.1 | 0.5 |
| 243 | Buan | 35°43′ | 126°42′ | 12.0 | 13.3 | 1.8 | 10.0 | 0.6 |
| 244 | Imsil | 35°36′ | 127°17′ | 247.9 | 248.7 | 1.7 | 10.0 | 0.6 |
| 245 | Jeongeup | 35°33′ | 126°51′ | 44.6 | 46.0 | 1.7 | 10.0 | 0.6 |
| 247 | Namwon | 35°24′ | 127°19′ | 90.3 | 94.7 | 1.8 | 10.0 | 0.6 |
| 248 | Jangsu | 35°39′ | 127°31′ | 406.5 | 408.3 | 1.6 | 10.0 | 0.6 |
| 260 | Jangheung | 34°41′ | 126°55′ | 45.0 | 45.3 | 1.9 | 10.2 | 0.5 |
| 261 | Haenam | 34°33′ | 126°34′ | 13.0 | 14.2 | 1.4 | 10.0 | 0.6 |
| 262 | Goheung | 34°37′ | 127°16′ | 53.1 | 54.4 | 1.6 | 10.0 | 0.6 |
| 271 | Bongwhoa | 36°56′ | 128°54′ | 319.8 | 322.3 | 1.6 | 10.0 | 0.6 |
| 272 | Yeongju | 36°52′ | 128°31′ | 210.8 | 211.7 | 1.5 | 10.0 | 0.5 |
| 273 | Mungyeong | 36°37′ | 128°08′ | 170.6 | 171.8 | 1.5 | 10.0 | 0.6 |
| 277 | Yeongdeok | 36°31′ | 129°24′ | 42.1 | 43.5 | 1.6 | 10.0 | 0.6 |
| 278 | Uiseong | 36°21′ | 128°41′ | 81.8 | 84.0 | 1.5 | 10.0 | 0.6 |
| 279 | Gumi | 36°07′ | 128°19′ | 48.9 | 48.9 | 1.5 | 10.0 | 0.6 |
| 281 | Yeongcheon | 35°58′ | 128°57′ | 93.6 | 94.5 | 1.7 | 10.0 | 0.5 |
| 284 | Geochang | 35°40′ | 127°54′ | 226.0 | 227.2 | 1.5 | 10.0 | 0.5 |
| 285 | Hapcheon | 35°33′ | 128°10′ | 33.1 | 34.1 | 1.5 | 10.0 | 0.6 |
| 288 | Miryang | 35°29′ | 128°44′ | 11.2 | 12.1 | 1.5 | 10.0 | 0.5 |
| 289 | Sancheong | 35°24′ | 127°52′ | 138.1 | 139.4 | 1.5 | 10.0 | 0.6 |
| 294 | Geoje | 34°53′ | 128°36′ | 46.3 | 47.5 | 1.5 | 10.0 | 0.5 |
| 295 | Namhae | 34°48′ | 127°55′ | 45.0 | 46.2 | 1.8 | 10.0 | 0.7 |

H: Height of observation field above mean sea level; Hb: Height of barometer above mean sea level; Ht: Height of thermometer above ground; Ha: Height of anemometer above ground; Hr: Height of raingauge above ground.

## Appendix B

**Table A2.** List of butterflies investigated in this study with their scientific names and distribution.

| Family | Scientific Name | | Distribution Pattern |
|---|---|---|---|
| Papilionidae | Parnassius stubbendorfii | Menetries, 1849 | Northern |
| | Parnassius bremeri | Bremer, 1864 | Northern |
| | Luehdor fiapuziloi | Erschoff, 1872 | Northern |
| | Sericinus montela | Gray, 1852 | Miscellaneous |
| | Byasa alcinous | Klug, 1836 | Miscellaneous |
| | Graphium sarpedon | Linnaeus, 1758 | Southern |
| | Papilioxuthus | Linnaeus, 1767 | Miscellaneous |
| | Papilio machaon | Linnaeus, 1758 | Miscellaneous |
| | Papilio memnon | Linnaeus, 1758 | Southern |
| | Papilio helenus | Linnaeus, 1758 | Southern |
| | Papilio protenor | Cramer, 1775 | Southern |
| | Papilio macilentus | Janson, 1877 | Southern |
| | Papilio bianor | Cramer, 1778 | Miscellaneous |
| | Papilio maackii | Menetries, 1858 | Miscellaneous |
| Pieridae | Leptidea amurensis | Menetries, 1859 | Northern |
| | Leptidea morsei | Fenton, 1882 | Northern |
| | Aporia crataegi | Linnaeus, 1758 | Northern |
| | Artogeia napi | Linnaeus, 1758 | Northern |
| | Pieris melete | Menetries, 1857 | Miscellaneous |
| | Artogeia canidia | Sparrman, 1768 | Miscellaneous |
| | Pieris rapae | Linnaeus, 1758 | Miscellaneous |
| | Pontia daplidice | Linnaeus, 1758 | Miscellaneous |
| | Anthocharis scolymus | Bulter, 1866 | Miscellaneous |
| | Gonepteryx maxima | Bulter, 1885 | Northern |
| | Gonepteryx aspasia | Menetries, 1858 | Miscellaneous |
| | Catopsilia pomona | Fabricius, 1755 | Southern |
| | Eurema mandarina | de l'Orza, 1869 | Southern |
| | Eurema laeta | Boisduval, 1836 | Southern |
| | Eurema brigitta | Stoll, 1780 | Southern |
| | Colias erate | Esper, 1805 | Miscellaneous |
| Lycaenidae | Curetis acuta | Moore, 1877 | Southern |
| | Taraka hamada | H.Druce, 1875 | Southern |
| | Spindasis takanonis | Matsumura, 1906 | Northern |
| | Arhopala japonica | Murray, 1875 | Southern |
| | Arhopala bazalus | Hewitson, 1862 | Southern |
| | Artopoetes pryeri | Murray, 1873 | Northern |
| | Coreana raphaelis | Oberthur, 1880 | Northern |
| | Ussuriana michaelis | Oberthur, 1880 | Northern |

**Table A2.** *Cont*.

| Family | Scientific Name | | Distribution Pattern |
|---|---|---|---|
| | Shirozua jonasi | Janson, 1877 | Northern |
| | Thecla betulae | Linnaeus, 1758 | Northern |
| | Protantigius superans | Oberthur, 1914 | Northern |
| | Japonica saepestriata | Hewitson, 1865 | Miscellaneous |
| | Jopnica lutea | Hewitson, 1865 | Miscellaneous |
| | Araragi enthea | Janson, 1877 | Northern |
| | Antigius attilia | Bremer, 1861 | Miscellaneous |
| | Antigius butleri | Fenton, 1882 | Northern |
| | Wagimo signata | Butler, 1881 | Northern |
| | Neozephyrus japonicus | Murray, 1875 | Northern |
| | Chrysozephyrus smaragdinus | Bremer, 1861 | Northern |
| | Chrysozephyrus brillantinus | Staudinger, 1887 | Northern |
| | Chrysozephyrus ataxus | Westwood, 1851 | Southern |
| | Favonius orientalis | Murray, 1875 | Northern |
| | Favonius korshunovi | Dubatolov et Sergeev, 1982 | Northern |
| | Favonius koreanus | Kim, 2006 | Northern |
| | Favonius ultramarinus | Fixsen, 1887 | Northern |
| | Favonius cognatus | Staudinger, 1892 | Northern |
| | Favonius taxila | Bremer, 1861 | Northern |
| Lycaenidae | Favonius yuasai | Shirozu, 1947 | Northern |
| | Favonius saphirinus | Staudinger, 1887 | Northern |
| | Satyrium herzi | Fixsen, 1887 | Northern |
| | Satyrium pruni | Linnaeus, 1758 | Northern |
| | Satyrium prunoides | Staudinger, 1887 | Northern |
| | Satyrium eximius | Fixsen, 1887 | Northern |
| | Satyrium latior | Fixsen, 1887 | Northern |
| | Satyrium walbum | Knoch, 1782 | Northern |
| | Callophrys ferrea | Butler, 1866 | Southern |
| | Callophrys frivaldszkyi | Kindermann, 1853 | Northern |
| | Rapala caerulea | Bremer et Grey, 1853 | Miscellaneous |
| | Rapala arata | Bremer, 1861 | Miscellaneous |
| | Lycaena dispar | Haworth, 1803 | Northern |
| | Lycaena phlaeas | Linnaeus, 1761 | Miscellaneous |
| | Niphanda fusca | Bremer et Grey, 1853 | Miscellaneous |
| | Chilades pandava | Horsfield, 1829 | Southern |
| | Jamides bochus | Stoll, 1782 | Southern |
| | Lampides boeticus | Linnaeus, 1767 | Southern |
| | Zizeeria maha | Kollar, 1844 | Southern |
| | Zizina otis | Fabricius, 1787 | Southern |
| | Cupido argiades | Pallas, 1771 | Miscellaneous |

**Table A2.** *Cont*.

| Family | Scientific Name | | Distribution Pattern |
|---|---|---|---|
| | Tongeia fischeri | Eversmann, 1843 | Miscellaneous |
| | Udara albocaerulea | Moore, 1879 | Southern |
| | Udara dilectus | Moore, 1879 | Southern |
| | Celastrina argiolus | Linnaeus, 1758 | Miscellaneous |
| | Celastrina sugitanii | Matsumura, 1919 | Northern |
| | Celastrina oreas | Leech, 1893 | Northern |
| Lycaenidae | Scolitantides orion | Pallas, 1771 | Northern |
| | Shijimiaeoidesdivina | Fixsen, 1887 | Northern |
| | Maculinea arionides | Staudinger, 1887 | Northern |
| | Maculinea teleius | Bergstrasser, 1779 | Northern |
| | Maculinea kurentzovi Sibatani | Hirowatari, 1994 | Northern |
| | Plebejus argus | Linnaeus, 1758 | Northern |
| | Plebejus argyrognomon | Bergstrasser, 1779 | Miscellaneous |
| | Plebejus subsolanus | Eversmann, 1851 | Northern |
| | Lybythea lepita | Moore, 1858 | Southern |
| | Parantica sita | Kollar, 1844 | Southern |
| | Parantica melaneus | Cramer, 1755 | Southern |
| | Danaus genutia | Cramer, 1779 | Southern |
| | Danaus chrysippus | Linnaeus, 1758 | Southern |
| | Melanitis leda | Linnaeus, 1758 | Southern |
| | Melanitis phedima | Cramer, 1780 | Southern |
| | Coenonympha amaryllis | Stoll, 1782 | Miscellaneous |
| | Coenonympha hero | Linnaeus, 1761 | Miscellaneous |
| | Coenonympha oedippus | Fabricius, 1787 | Northern |
| | Lopinga achine | Scopoli, 1763 | Miscellaneous |
| | Lasiommata deidamia | Eversmann, 1851 | Miscellaneous |
| Nymphalidae | Kirinia epimenides | Menetries, 1859 | Northern |
| | Kirinia epimenidas | Staudinger, 1887 | Northern |
| | Mycalesis francisca | Stoll, 1780 | Southern |
| | Mycalesis gotama | Moore, 1858 | Southern |
| | Lethe marginalis | Motschulsky, 1860 | Miscellaneous |
| | Lethe diana | Butler, 1866 | Miscellaneous |
| | Ninguta schrenckii | Menetries, 1858 | Northern |
| | Aphantopus hyperantus | Linnaues, 1758 | Northern |
| | Melanargia halimede | Menetries, 1858 | Miscellaneous |
| | Melanargia epimede | Staudinger, 1887 | Northern |
| | Oeneisurda | Eversmann, 1847 | Northern |
| | Oeneis mongolica | Oberthur, 1876 | Northern |
| | Minois dryas | Scopoli, 1763 | Northern |

**Table A2.** *Cont*.

| Family | Scientific Name | | Distribution Pattern |
|---|---|---|---|
| | Eumenis autonoe | Esper, 1783 | Northern |
| | Ypthima argus | Butler, 1866 | Miscellaneous |
| | Ypthima multistriata | Butler, 1883 | Miscellaneous |
| | Ypthima motschulskyi | Bremer et Grey, 1853 | Miscellaneous |
| | Erebia cyclopius | Eversmann,1844 | Northern |
| | Erebia wanga | Bremer, 1864 | Northern |
| | Argynnis paphia | Linnaues, 1758 | Miscellaneous |
| | Argynnis childreni | Gray, 1831 | Miscellaneous |
| | Argynnis zenobia | Leech, 1890 | Northern |
| | Argynnis sagana | Doubleday, 1847 | Miscellaneous |
| | Argynnis laodice | Pallas, 1771 | Miscellaneous |
| | Argynnis ruslana | Motschulsky, 1866 | Miscellaneous |
| | Argynnis anadyomene | C. et R. Felder, 1862 | Northern |
| | Argynnis niobe | Linnaeus, 1758 | Miscellaneous |
| | Argynnis vorax | Butler, 1871 | Miscellaneous |
| | Argynnis nerippe | Felder,1862 | Miscellaneous |
| | Argynnis aglaja | Linnaeus, 1758 | Northern |
| | Argyreus hyperbius | Linnaeus, 1763 | Southern |
| | Brenthis daphne | Bergstrasser, 1780 | Northern |
| | Brenthis ino | Rottemburg, 1775 | Northern |
| Nymphalidae | Boloria thore | Hubner, 1803–1804 | Northern |
| | Boloria oscarus | Eversmann,1844 | Northern |
| | Boloria perryi | Butler, 1882 | Northern |
| | Boloria selene | Schiffermuller, 1775 | Northern |
| | Limenitis camilla | Linnaeus, 1764 | Miscellaneous |
| | Limenitis doerriesi | Staudinger, 1892 | Northern |
| | Limenitis helmanni | Lederer, 1853 | Northern |
| | Limenitis homeyeri | Tancre, 1881 | Northern |
| | Limenitis sydyi | Lederer, 1853 | Northern |
| | Limenitis amphyssa | Menetries, 1859 | Northern |
| | Limenitis moltrechti | Kardakoff,1928 | Northern |
| | Limenitis populi | Linnaeus, 1758 | Northern |
| | Seokia pratti | Leech, 1890 | Northern |
| | Neptis sappho | Pallas, 1771 | Miscellaneous |
| | Neptis philyra | Menetries, 1858 | Northern |
| | Neptis philyra | Staudinger, 1887 | Northern |
| | Neptis speyeri | Staudinger, 1887 | Northern |
| | Neptis rivularis | Scopoli, 1763 | Northern |
| | Neptis pryeri | Butler, 1871 | Miscellaneous |
| | Neptis andetria | Fruhstorfer, 1912 | Northern |

**Table A2.** *Cont*.

| Family | Scientific Name | | Distribution Pattern |
|---|---|---|---|
| | Neptis alwina | Bremer et Grey, 1853 | Miscellaneous |
| | Neptisthisbe | Menetries, 1859 | Northern |
| | Neptis tshetverikovi | Kurentzov, 1936 | Northern |
| | Neptisilos | Fruhstorfer, 1909 | Northern |
| | Neptis raddei | Bremer, 1861 | Northern |
| | Dichorragia nesimachus | Doyere, 1840 | Southern |
| | Apatura ilia | Schiffermuller, 1775 | Northern |
| | Apaturametis | Freyer, 1829 | Northern |
| | Apatura iris | Linnaeus, 1758 | Northern |
| | Mimathyma schrenckii | Menetries, 1859 | Northern |
| | Mimathyma nycteis | Menetries, 1859 | Northern |
| | Chitoriaulupi | Doherty, 1889 | Miscellaneous |
| | Dilipafenestra | Leech, 1891 | Miscellaneous |
| | Hestina persimilis | Westwood, 1850 | Southern |
| | Hestina assimilis | Linnaeus, 1758 | Southern |
| | Sasakiacharonda | Hewitson, 1863 | Miscellaneous |
| | Sephisa princeps | Fixsen, 1887 | Miscellaneous |
| | Cyrestis thyodamas | Doyere, 1840 | Southern |
| Nymphalidae | Araschnia levana | Linnaeus, 1758 | Northern |
| | Araschnia burejana | Bremer, 1861 | Northern |
| | Vanessa cardui | Linnaeus, 1758 | Miscellaneous |
| | Vanessa indica | Herbst, 1794 | Miscellaneous |
| | Polygonia c–aureum | Linnaeus, 1758 | Miscellaneous |
| | Polygonia c–album | Linnaeus, 1758 | Northern |
| | Nymphalis l–album | Esper, 1780 | Northern |
| | Nymphalis xanthomelas | Esper, 1781 | Northern |
| | Nymphalis antiopa | Linnaeus, 1758 | Northern |
| | Aglais urticae | Linnaeus, 1758 | Northern |
| | Aglias io | Linnaeus, 1758 | Northern |
| | Kaniska canace | Linnaeus, 1763 | Miscellaneous |
| | Junonia almanda | Linnaeus, 1758 | Southern |
| | Junonia orithya | Linnaeus, 1758 | Southern |
| | Hypolimnas misippus | Linnaeus, 1764 | Southern |
| | Hypolimnas bolina | Linnaeus, 1758 | Southern |
| | Euphydryas davidi | Oberthur, 1881 | Northern |
| | Melitaeaambigua | Menetries, 1859 | Northern |
| | Melitaea britomartis | Assmann, 1847 | Northern |
| | Melitaea protomedia | Menetries, 1858 | Northern |
| | Melitaea scotosia | Butler, 1878 | Northern |

**Table A2.** *Cont.*

| Family | Scientific Name | | Distribution Pattern |
|---|---|---|---|
| Hesperiidae | Choaspes benjaminii | Guerin–Meneville, 1843 | Southern |
| | Burara aquilina | Speyer, 1879 | Northern |
| | Burara striata | Hewitson, 1867 | Southern |
| | Lobocla bifasciata | Bremer et Grey, 1853 | Miscellaneous |
| | Satarupa nymphalis | Speyer, 1879 | Northern |
| | Daimio tethys | Menetries, 1857 | Miscellaneous |
| | Erynnis montanus | Bremer, 1861 | Miscellaneous |
| | Pyrgus maculatus | Bremer et Grey, 1853 | Miscellaneous |
| | Pyrgus malvae | Linnaeus, 1758 | Northern |
| | Cartero cephalus | Graeser, 1888 | Northern |
| | Cartero cephalus silvicola | Meigen, 1828 | Northern |
| | Heteropterus morpheus | Pallas, 1771 | Northern |
| | Leptalina unicolor | Bremer et Grey, 1853 | Miscellaneous |
| | Isoteinon lamprospilus | C. et R. Felder, 1862 | Southern |
| | Aeromachus inachus | Menetries, 1859 | Miscellaneous |
| | Thymelicus leoninus | Butler, 1878 | Miscellaneous |
| | Thymelicus sylvaticus | Bremer, 1861 | Miscellaneous |
| | Ochlodes similis | Leech, 1893 | Northern |
| | Ochlodes venatus | Bremer et Grey, 1853 | Miscellaneous |
| | Ochlodes ochraceus | Bremer, 1861 | Northern |
| | Ochlodes subhyalina | Bremer et Grey, 1853 | Miscellaneous |
| | Hesperia florinda | Butler, 1878 | Northern |
| Hesperiidae | Potanthus flavus | Murray, 1875 | Miscellaneous |
| | Polytremis zina | Evans, 1932 | Northern |
| | Pelopidas jansonis | Butler, 1878 | Southern |
| | Peolpidas siensis | Mabile, 1877 | Miscellaneous |
| | Peolpidas mathias | Fabricius, 1798 | Southern |
| | Parnara guttata | Bremer et Grey, 1853 | Southern |

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
