# Peer review of "Spatial Distribution of Butterflies in Accordance with Climate Change in the Korean Peninsula"

_sustainability, doi:10.3390/su12051995_

Round 1

Reviewer 1 Report

Thanks.

Author Response

Authors are most thankful to the reviewer for the comments and issues.

Reviewer 2 Report

The manuscript has improved much from the first version that I saw. The study is well described and shows interesting results. For me it is fine.

Author Response

The manuscript has improved much from the first version that I saw. The study is well described and shows interesting results. For me it is fine.

=> Authors are very much thankful for the comments and criticism toward the manuscript.

This manuscript is a resubmission of an earlier submission. The following is a list of the peer review reports and author responses from that submission.

Round 1

Reviewer 1 Report

Difficulties with the English sometimes interfere with a clear understanding of the text. Because habitats per se are not discussed or itemized, the repeated use of "habitats" to describe occupied areas is confusing and should be replaced by a term like "area of occupation." In paragraph 2.2, there was no such methodology as the Pollard-walk method before the 1970s, so older data could not have been obtained by such transect methods--but presumably reflect traditional distributional-data methods, mainly collection records. This must be clarified, and some evaluation made, even if subjective, of the quality of the data. (I realize that this information may be imbedded in references 13-16, but these are not available to me to check except by interlibrary loan, which would substantially delay this review. These references are not readily available globally!) If one assumes the quality of the data justifies the analysis, the conclusions are seemingly warranted. But in the absence of any discussion of the data quality, one simply cannot judge. For example, one does not know whether there has been any historic discrepancy in the intensity of collecting or whether recent transect sampling has been uniform. 

Author Response

Difficulties with the English sometimes interfere with a clear understanding of the text. Because habitats per se are not discussed or itemized, the repeated use of "habitats" to describe occupied areas is confusing and should be replaced by a term like "area of occupation."

=> We are sorry to make things confusing, but habitats mean cell grids based on the map. The long-term data of butterflies were reconstructed using 4 books that were published each time periods.

In paragraph 2.2, there was no such methodology as the Pollard-walk method before the 1970s, so older data could not have been obtained by such transect methods--but presumably reflect traditional distributional-data methods, mainly collection records. This must be clarified, and some evaluation made, even if subjective, of the quality of the data. (I realize that this information may be imbedded in references 13-16, but these are not available to me to check except by interlibrary loan, which would substantially delay this review. These references are not readily available globally!)

=> We obtained the data based on published books so that dataset were not available as you indicated; however, this kind of long term study (with standard field methods) should be able to provide trends of butterflies in the Korean peninsula with global warming implications.

If one assumes the quality of the data justifies the analysis, the conclusions are seemingly warranted. But in the absence of any discussion of the data quality, one simply cannot judge. For example, one does not know whether there has been any historic discrepancy in the intensity of collecting or whether recent transect sampling has been uniform.

=> The quality of data can be somewhat questionable, but the published books were done by government leading organization in experts with standardized methods over 1938-2011 so that we believed that it has meaningful information.

Reviewer 2 Report

The paper intended to answer the question that how temperature can impact the distribution of butterflies in the Korean Peninsula, and discussed the potential sequence of global warming on human beings.

However, the paper failed to describe the dataset and justify the method that was used. Some English writing is also confusing that hinders the understanding of the paper.

For example, without reading the full paper, it was hard to understand the statement in the abstract between line 17 and line 19.

A detailed description of the butterfly distribution data would be helpful, especially, a map showing the collected butterfly data.

Please add a reference and a short explanation of the Tukey HSD method.

Section 3.1 discussed the temperature changes over the years and across latitudes. I don't think it is necessary to prove that the temperature will change across latitudes.

Please provide a clear name for Figure 4.

What does the number of cell grids mean? The number of butterflies?

What do you mean by habitats change, the change of sizes?

Please provide more explanation of the Gradient Analysis, and why choose this analysis?

Why did you divide the study area based on latitudes rather than distances are areas? The linear distances and areas are different between 36-37 and 35-36 latitude regions.

Author Response

The paper intended to answer the question that how temperature can impact the distribution of butterflies in the Korean Peninsula, and discussed the potential sequence of global warming on human beings.

However, the paper failed to describe the dataset and justify the method that was used. Some English writing is also confusing that hinders the understanding of the paper.

=> English edting was done. Thank you.

1) For example, without reading the full paper, it was hard to understand the statement in the abstract between line 17 and line 19.

=> Changed sentences as “Southern butterflies (with northern limit) tend to increase in appearance between temperature and number of habitats (P<0.000), while northern butterflies (with southern limit) show no statistical significance between temperature and number of habitats, indicating their sensitivity to temperature change.” lines 17-19

2) A detailed description of the butterfly distribution data would be helpful, especially, a map showing the collected butterfly data.

=> We are sorry being unable to provide such data due to being constructed based on published books in government (not collecting original data).

3) Please add a reference and a short explanation of the Tukey HSD method.

=> changed as ‘Tukey HSD that is applicable to pairwise comparison of means was applied for post-hoc to monitor change of distribution during four time periods [17]. “ The reference was added.

4) Section 3.1 discussed the temperature changes over the years and across latitudes. I don't think it is necessary to prove that the temperature will change across latitudes.

=> Please understand that our assumptions for northern and southern species were based on latitude so that we need the temperature trends.

5) Please provide a clear name for Figure 4.

=> changed to “Temperature range of habitat with time periods in southern and northern species during 1938-2011.” Thank you for pointing out this.

6) What does the number of cell grids mean? The number of butterflies?

=> The grid cell was created on latitude 0.5° (56 km) x longitude 0.5° (44.4 km), and observation was marked in the grid cell, and a total of 99 grid cells were created (9 for longitude, 11 for latitude).

(lines 90-91) The grid cells to represent of the habitat that species were found.

7) What do you mean by habitats change, the change of sizes?

=> habitat changes mean that # of grid cells changed according to species presence due to global warming.

8) Please provide more explanation of the Gradient Analysis, and why choose this analysis?

=> gradient analysis seems to be inappropriate so that we deleted the paragraph. and table 3.

9) Why did you divide the study area based on latitudes rather than distances are areas? The linear distances and areas are different between 36-37 and 35-36 latitude regions.

=> The study was based on the cell grids where the species were present so that it was much better (and almost impossible using distance factor).

Round 2

Reviewer 1 Report

The authors' replies do NOT adequately respond to the criticisms. They merely state that they believe the data are credible. The text has not been modified to address the data-quality issue. It is not acceptable to use rigorous quantitative methods on questionable data.

Reviewer 2 Report

Thanks for the revision.

Probably a final round of English editing will help the overall quality of the paper.